# Genomic Characteristics and E Protein Bioinformatics Analysis of JEV Isolates from South China from 2011 to 2018

**DOI:** 10.3390/vaccines10081303

**Published:** 2022-08-12

**Authors:** Yawei Sun, Hongxing Ding, Feifan Zhao, Quanhui Yan, Yuwan Li, Xinni Niu, Weijun Zeng, Keke Wu, Bing Ling, Shuangqi Fan, Mingqiu Zhao, Lin Yi, Jinding Chen

**Affiliations:** 1Department of Microbiology and Immunology, College of Veterinary Medicine, South China Agricultural University, Guangzhou 510642, China; 2Guangdong Laboratory for Lingnan Modern Agriculture, Guangzhou 510642, China; 3Key Laboratory of Zoonosis Prevention and Control of Guangdong Province, Guangzhou 510642, China

**Keywords:** Japanese encephalitis virus, genomic characteristics, E protein, bioinformatics

## Abstract

Japanese encephalitis is a mosquito-borne zoonotic epidemic caused by the Japanese encephalitis virus (JEV). JEV is not only the leading cause of Asian viral encephalitis, but also one of the leading causes of viral encephalitis worldwide. To understand the genetic evolution and E protein characteristics of JEV, 263 suspected porcine JE samples collected from South China from 2011 to 2018 were inspected. It was found that 78 aborted porcine fetuses were JEV-nucleic-acid-positive, with a positive rate of 29.7%. Furthermore, four JEV variants were isolated from JEV-nucleic-acid-positive materials, namely, CH/GD2011/2011, CH/GD2014/2014, CH/GD2015/2015, and CH/GD2018/2018. The cell culture and virus titer determination of four JEV isolates showed that four JEV isolates could proliferate stably in Vero cells, and the virus titer was as high as 10^8.5^ TCID 50/mL. The whole-genome sequences of four JEV isolates were sequenced. Based on the phylogenetic analysis of the JEV E gene and whole genome, it was found that CH/GD2011/2011 and CH/GD2015/2015 belonged to the GIII type, while CH/GD2014/2014 and CH/GD2018/2018 belonged to the GI type, which was significantly different from that of the JEV classical strain CH/BJ-1/1995. Bioinformatics tools were used to analyze the E protein phosphorylation site, glycosylation site, B cell antigen epitope, and modeled 3D structures of E protein in four JEV isolates. The analysis of the prevalence of JEV and the biological function of E protein can provide a theoretical basis for the prevention and control of JEV and the design of antiviral drugs.

## 1. Introduction

Japanese encephalitis (JE) is an acute zoonotic disease caused by the Japanese encephalitis virus (Japanese encephalitis virus, JEV) [1,2]. JEV is a member of the genus Favivirus of the arboviridae (Faviviridae). Pigs are the primary hosts and source of infection of JEV, and other domestic animals such as sheep, cattle, horses, mules, and dogs are other primary sources of infection or act as storage hosts. JEV causes significant economic losses to the pig industry, but also causes severe viral encephalitis in human beings [3]. It is reported that the fatality rate of JEV infection can reach 30%, and nearly half of the survivors are reported to have permanent neuropsychiatric sequelae [4,5]. JEV has long been considered the leading cause of Asian viral encephalitis, of which about 50 percent occurs in China. The occurrence of JE is closely related to the environmental climate (temperature, humidity) and mosquito density, and it is most common in rainy areas [6]. Compared with the north, this is also an important reason why the distribution of Japanese encephalitis is concentrated in the south of China [7].

The genome of JEV is approximately 11 kb in length and consists of untranslated regions (UTRs), 30 utrs, and a single open reading frame (ORF). The JEV genome encodes in the order 5 ‘-C-PrM-E-NS1-(NS2A, B)-NS3-(NS4A, B)-NS5-3’ from the 5 ‘end to the 3’ end. The ORF is approximately 10.3 kb in size and encodes structural (C, PrM/M, E) and nonstructural proteins (NS1, NS2A, NS2B, NS3, NS4A, NS4B, and NS5) [8,9]. According to the phylogenetic analysis of the nucleotide sequence of the E gene, JEV can be classified into five genotypes, i.e., GI, GII, GIII, GIV, and GV [1,10]. Studies have shown that JEV GI can be further divided into two subgenotypes: JEV GIa and JEV GIb. Currently, most of the isolates of JEV from mainland China belong to JEV GIb. GIb gradually replaced GIII in the 1990s, even though it had been an endemic area of JEV GIII, including in Korea and Japan [11,12]. Meanwhile, studies confirm that JEV GIb can cause human JE outbreaks [13].

Although the C protein and nonstructural proteins are also involved in the virulence control of the Japanese encephalitis virus to some extent, the critical determinants of JEV virulence are the PrM and E proteins [14,15]. In general, flavivirus E protein exists as an antiparallel dimer on the surface of mature virions. The fusion peptide of DII snuggles in the cavity composed of DI and DIII on the relative subunit. Unlike other flaviviruses, JEV dimer is mainly maintained by fusion ring–fusion ring pocket interaction and lacks DII-DII contact [16,17]. The JEV E gene encodes the viral sacsin protein, which consists of 411 amino acid residues, and the three-dimensional structure of sacsin is mainly composed of three domains, with the main domain I (DI) being E-1-E-51, E-137-E-196, and E-293-E-311. Extended domain II (DII) is E-52-E-137 and E-197-E-292. Additionally, E310-E411 is the globular domain III (DIII). DI consists of nine β chains between DII and DIII, DII consists of two extended loops protruding from DI, and DIII has an immunoglobulin-like fold and is located at the C-terminus of the exon, connected to DI by a short peptide [17]. The E protein is the major protein that brings the virus into the cell in fusion with the cell membrane in association with cell membrane receptors and can induce the production of neutralizing antibodies [18,19]. The DII region folds into an immunoglobulin-like structure on the viral surface. Many antigens neutralizing epitopes concentrate on this domain, are involved in receptor binding, are an essential antigenic determinant, and have both hemagglutinating and neutralizing activities [20]. Specific amino acid mutations of E protein have been found to be crucial for the virulence of the virus [21]. Monitoring the amino acid residues and mutations of E protein, including E-107, E-138, E-123, E-176, E-177, E-244, E-279, etc., has provided a theoretical basis for elucidating the mechanism of the safety of JEV-attenuating vaccines and contributed to the development of new vaccines and the quality control of new vaccines [22].

JEV monitoring and analysis are essential to prevent and control pig JE effectively. Therefore, the isolation of suspected JE from pig farms in South China is carried out in this study. The isolates’ genomic characteristics and E protein bioinformatics are studied to provide theoretical support for JE prevention and control.

## 2. Materials and Methods

### 2.1. Viruses and Cells

The vaccine strain CH/SA14-14-2 MSV/2018 (Genbank: MH258849.1), Vero cells and JEV polyclonal antibody used in the study were preserved by the Department of Microbiology and Immunology, School of Veterinary Medicine, South China Agricultural University.

### 2.2. Identification of Clinical Materials and Virus Isolation

From 2011 to 2018, 263 samples of pigs suspected of JE were collected from several pig farms in Guangdong, Guangxi and Hainan provinces. The diseased materials were placed in a precooled milk bowl. Next, 3.0 mL serum-free MEM cell culture medium was added, ground and mixed repeatedly, then frozen and thawed three times. The supernatant was centrifuged and filtered with a 0.22 μM filter to remove bacteria and stored at −80 °C. According to the instructions of the Takara virus DNA/RNA extraction kit, virus RNA was extracted, and JEV cDNA synthesis was carried out with the instructions of the PrimeScript virus II1st Strand cDNA SynthesisKit kit. According to the sequence of the JEV genome in GenBank (MH258849.1), one pair of identification primers (JEV F1:5′-ACAGTCCGTTGTTGCTCTTG-3′ and JEV R1:5′-GGTGGTTGATCTGCTTGTC-3′) and four pairs of primers designed by PrimePremier5.0 (Table 1) were used for the amplification of JEV genome. The DNA reactants of JEV were purified according to the instructions of the JEV gel purification kit made by the OMEGA Company. The purified product was sent to Shanghai Bioengineering Co., Ltd. for sequencing. The positive samples were filtered to remove bacteria, inoculated to grow into monolayer Vero cells, and incubated at 37 °C for one h. The cell supernatant was discarded; then, the samples were supplemented with 2% FBS MEM at 37 °C to continue the culture, and were observed day by day. When the cells showed pathological changes, the cell cultures were harvested in time and identified by RT-PCR. If there were no pathological changes, the cell cultures were gathered on the 5th day of culture, frozen and thawed repeatedly three times and stored at −80 °C.

### 2.3. Serological Identification, Subculture and Determination of Virus Content

The cytopathic change culture identified by PCR was mixed with JEV polyclonal antibody and incubated at 37 °C for one h; then, Vero cells were inoculated and cultured at 37 °C 5% CO_2_. A control group was set up and observed day by day. After plaque purification, the JEV-positive isolates were diluted with a serum-free MEM culture medium ten times. The diluted virus solution was inoculated into 96-well plate Vero cells. After incubation for 2 h, the supernatant was discarded. After cleaning with MEM cell culture medium, 100 μL of MEM culture medium containing 2% FBS was added to each well, and a cell blank control group was set up. Then, 5% CO2 was cultured at 37 °C for seven days and observed daily. The number of cytopathic pores was recorded, and the virus titer of porcine JEV was calculated by the Reed–Muench method [23].

### 2.4. Analysis of Genomic Characteristics and Genetic Variation of Amino Acids

To understand the genetic evolution characteristics of the whole genome of JEV isolates, the sequences of 40 domestic and foreign JEV strains and classical strains were collected from NCBI (https://www.ncbi.nlm.nih.gov/) (accessed on 20 April 2021) (Appendix A). The obtained gene sequences were spliced with the SeqMan program on DNAstar software (Headquarters: Madison, WI, USA), and the spliced sequences and reference strains were compared with MAFFT software (Research Institute for Microbial Diseases, Osaka University, Suita, Japan) [24]. Then, the genomes and E genes were analyzed by SDTv1.2. We used ClustalW v2.0 (Kyoto University Bioinformatics Center, Kyoto, Japan) to compare nucleotides and deduce amino acid sequences, and edited them manually using MEGA X (Department of Biological Sciences, Tokyo Metropolitan University, Hachioji, Japan) [25]. Using the maximum compound likelihood nucleotide substitution model realized by the adjacent join (NJ) method in MEGAX, the phylogenetic analysis of the complete coding region of nucleotide sequences (nt) and each protein gene data set was carried out. One thousand bootstrap copies were used to evaluate the robustness of the system diagram.

### 2.5. Bioinformatics Analysis

The N-glycosylation sites of JEV isolates were predicted by NetNGlyc4.0Server (https://www.cbs.dtu.dk/services/NetNGlyc/) (accessed on 20 February 2022), and the phosphorylation sites of the proteins of JEV isolates and reference strains were indicated by NetPhos 3.1Server (https://www.cbs.dtu.dk/services/NetPhos/) (accessed on 20 February 2022). The T cell epitopes of JEV isolates and vaccine reference strains were predicted by the IEDB website (http://www.iedb.org/) (accessed on 22 February 2022) using SYFPEITHI, the database of MHC ligands and peptide motifs, and the antigenicity and B cell epitopes of JEV isolates were analyzed by Protean tool on DNAstar software and ABCpred prediction (http://crdd.Osdd.Net/raghava/abcpred/ABC_submission.Html) (accessed on 22 February 2022). The amino acid sequences of E protein of four JEV isolates and the vaccine strain CH/SA14-14-2 MSV/2018 were compared by a blast to select the optimal template PDB number (more than 60% homology). Then, Modeller software (version 10.3, California Institute for Quantitative Biomedical Research, USA)was used for multi-template modeling [26], and finally, this was visually displayed by PyMol software (Company: DeLano Scientific LLC, San Carlos, CA, USA) [27].

## 3. Results

### 3.1. Clinical Sample Detection and Virus Isolation and Identification

From 2011 to 2018, 263 samples were collected from 14 pig farms in the Guangdong and Guangxi provinces for JEV testing (Figure 1). Among the 263 samples, 78 were positive for the porcine JEV nucleic acid test, with a positive rate of 29.7% (78/263). After the positive samples were treated and used for virus isolation on Vero cells, four JEV isolates were obtained, namely, CH/GD2011/2011, CH/GD2014/2014, CH/GD2015/2015, and CH/GD2018/2018. Four strains of cDNA were used as templates for PCR identification with JEV-specific primers, and the results were amplified bands of about 437bp (Figure 2). Variants CH/GD2011/2011, CH/GD2014/2014, CH/GD2015/2015, and CH/GD2018/2018 were identified by porcine Japanese-encephalitis-specific positive serum. The results showed that CPE was not found in the neutralization group, serum control group, or the cell control group of the four JEV isolates, while CPE was found in the virus control group. The pathological changes were characterized by aggregation, enlargement, separation, and syncytial formation (Figure 3). The results showed that all four isolates could be neutralized by the specific serum of porcine Japanese encephalitis, and all four isolates were porcine JEV. Four porcine JEV isolates were continuously cultured on Vero cells for four generations, and the F4 generation of the isolated strain was cloned and purified three times. The virus content of the purified strain was continuously passaged on Vero cells for the third time (Appendix A). The results showed that when all the isolated strains were transferred to F10 generation, the virus propagated stably, and the virus content was 10^7.5^~10^8.7^ TCID 50/mL. Using four JEV isolates of cDNA as the template and ddH_2_O as the control, PCR amplification was carried out using JEV whole-gene amplification primers (Appendix A). The corresponding bands of four isolates were amplified by JEV complete gene amplification primers, consistent with the expected size. All the amplified products were purified, sent to Shanghai Bioengineering Technology Service Co., Ltd. for gene sequencing, and then spliced with SeqMan.

### 3.2. Nucleotide and Amino Acid Similarity and Genetic Evolution Analysis of JEV Isolates

The genome and E gene sequences of JEV isolates and different genotype reference strains were analyzed by the MAFFT algorithm in MAFFT software, and the genomic phylogeny was analyzed by MEGA X software. Finally, the NJ phylogenetic tree based on the JEV genome and E gene was obtained (Figure 4). The results showed that the four isolates and reference strains were divided into five clusters regarding genomic and genetic evolution: GI, GII, GIII, GIV and GV. Variants CH/GD2011/2011 and CH/GD2015/2015 were located in the GIII cluster, and their genetic relationships were close to the CH/SC2016/2016 strain and the CH/SA14-14-2 MSV/2018 strain. Strains CH/GD2014/2014 and CH/GD2018/2018 were located in the GI cluster. Their genetic relationship was closer to the CH/SD0810/2008 strain and the JPN/Mo-Kagawa/2020 strain. Four JEV isolates were more closely related to the GII, GIV and GV JEV classical strains (Figure 4a). The NJ phylogenetic tree based on the E gene of four JEV isolates and reference strains showed that CH/GD2011/2011 and CH/GD2015/2015 variants were also located in the GIII cluster. The genetic relationship was closer to the CH/Hubei/2015 strain and CH/SA14-14-2 MSV/2018 strain, while CH/GD2014/2014 and CH/GD2018/2018 variants were also located in the GI cluster. Their genetic relationship was closer to the CH/SD0810/2008 strain and CH/anheal/2017 strain (Figure 4b).

The genomic nucleotides and deduced amino acid sequences of four JEV isolates and GI, GII, GIII, GIV and GV JEV classical strains were analyzed. The results showed that the nucleotide similarity and deduced amino acid sequence similarity of the E gene of the four JEV isolates and reference strains were 76.2~99.9% and 89.6~100.0%, respectively. In contrast, the genomic nucleotide sequence similarity was 77.9~99.2%. (Figure 5). At the same time, the nucleotides and deduced amino acid sequences of C, PrM, NS1, NS2a, NS2b, NS3, NS4a, NS4b and NS5 genes of the isolate were 75.8%~100.0% and 78.9%~100.0%, respectively (Appendix A).

The protein encoded by the JEV E gene is the main structural protein and antigen protein of JEV which contains the core antigen determinant and is closely related to the virus’ adsorption, entry and pathogenicity [28]. The amino acid sequence analysis of E protein of four JEV isolates and reference strains showed that both CH/GD2011/2011 and CH/GD2015/2015 variants had the same amino acid mutations at positions 222, 327 and 366, which were S (serine) to A (alanine), T (threonine) to S (serine), and S (serine) to A (alanine), respectively. The mutation characteristics were similar to those of classical GIII strains such as CH/SA14-14-2/2018, indicating that the above amino acid sites were high-frequency mutations in the E protein of JEV GIII strains. Compared with the reference strain, the CH/GD2014/2014 variant and the CH/GD2018/2018 variant had no amino acid mutation, which indicated that the amino acid sequence of E protein of the JEV GI strain was higher than that of the JEV GI strain. The amino acid sequence of the E protein of the GIII strain was prone to mutation. The 107th amino acid of the E protein of the CH/GD2011/2011 variant mutated from L (leucine) to F (phenylalanine), the 176th amino acid from I (isoleucine) to V (valine), the 244th amino acid from E (glutamic acid) to G (glycine), and the 279th amino acid from K (lysine) to M (methionine). It is suggested that neurotoxicity may be weakened by the mutation of the above amino acid sites (Figure 6).

### 3.3. Bioinformatics Analysis of E Gene

#### 3.3.1. Glycosylation Site Analysis

Glycosylation modification is a significant post-translational modification. Glycosylation modification affects the spatial conformation, activity, transport and localization of proteins and plays a vital role in signal transduction, molecular recognition and immunity. At the same time, glycosylation plays an essential role in virus replication, infection, immunity and virulence [29,30,31]. The N-glycosylation sites of the proteins encoded by the E gene of four JEV isolates were analyzed by NetNGlyc1.0Server. All four JEV E proteins had a potential N-glycosylation site at N154, and the confidence level reached 9/9. The glycosylation sites of E protein of the four JEV isolates were the same as the potential glycosylation sites of the CH/SA14-14-2MSV/2018 protein of the CH/BJ-1/1995 strain and the vaccine strain of JEV. The potential glycosylation sites of JEVE protein did not change (the results are not shown), suggesting that the glycosylation sites of the JEVE protein may be conserved in different genotypes and strains.

#### 3.3.2. Phosphorylation Site Analysis

Protein phosphorylation is a type of post-translational modification of proteins. Phosphorylation is mainly concentrated on tyrosine, threonine, and serine residues, and an additional charge is added to the protein after phosphorylation. At the same time, the unique size and charge characteristics of covalently linked phosphates also allow phosphorylated proteins to specifically recognize phosphorylated proteins through phosphorylation-specific binding domains in other proteins, thus promoting inducible protein–protein interactions [32]. Biological software deduced the E gene sequences of four JEV isolates into amino acid sequences. The phosphorylation sites of E protein of four JEV isolates were predicted by NetPhos3.1Server. It was found that E protein had 72~79 potential phosphorylation sites. There are 72 potential phosphorylation sites of E protein in the CH/GD2014/2014 variant, including 27 threonine phosphorylation sites, 36 serine phosphorylation sites, and 9 tyrosine phosphorylation sites, and 79 potential phosphorylation sites in the CH/GD2015/2015 variant, including 31 threonine phosphorylation sites, 38 serine phosphorylation sites, and 10 tyrosine phosphorylation sites. There are 76 potential phosphorylation sites in the E protein of the CH/GD2014/2014 variant, including 31 threonine phosphorylation sites, 36 serine phosphorylation sites, and 9 tyrosine phosphorylation sites in the CH/GD2014/2014 variant, and 76 potential phosphorylation sites in the CH/GD2018/2018 variant, including 31 threonine phosphorylation sites, 36 serine phosphorylation sites, and 9 tyrosine phosphorylation sites (Appendix A). These phosphorylation sites suggest that the E protein of the JEV isolate may play an essential role in virus replication and immune regulation.

#### 3.3.3. Analysis of Antigenic Epitopes of T/B Cells

Using the Protean tool of DNAstar software and ABCpredPrediction online software to analyze the B cell epitopes of the proteins encoded by four JEV isolates can provide a theoretical basis for selecting vaccine candidates and vaccine development. According to the predicted results of ABCpredPrediction, a confidence value greater than 0.85 was used as the standard of the B cell epitope. The results showed that CH/GD2011/2011, CH/GD2015/2015, CH/GD2014/2014, CH/GD2018/2018 and JEV CH/SA14-14-2 MSV/2018 and CH/BJ-1/1995 had potential B cell epitopes at positions 24~39, 183~198, 332~347, 418~433 and 455~470, respectively. The confidence levels were 0.91, 0.95, 0.91, 0.90 and 0.97, respectively. Both CH/GD2014/2014 and CH/GD2018/2018 variants had a potential B cell epitope at positions 221~236, with a confidence level of 0.86 (Appendix A). The results showed that the E protein of four JEV isolates had good antigenicity and could stimulate the body to induce a humoral immune response (Figure 7).

T cell epitopes refer to antigenic epitopes that can be recognized by T cell receptors and induce cellular immune responses. Using the existing database of MHC ligands and polypeptide motif SYFPEITHI, the critical value of the software prediction score was set to 25 to predict the T cell epitopes of the virus proteins of the four JEV isolates and vaccine strains, respectively. Prediction results: JEV CH/GD2011/2011 variant E protein may contain 11 potential candidate T cell epitopes; JEV CH/GD2015/2015 variant E protein may contain 12 potential candidate T cell epitopes; the JEV CH/GD2014/2014 and JEV CH/GD2018/2018 variants may contain 10 potential candidate T cell epitopes and 10 repeats with the JEV CH/GD2011/2011 variant; and the CH/GD2015/2015 variant has one more “GSQEGGLHLALAGAI” T cell epitope (Table 2).

#### 3.3.4. Modeled 3D Structure Analysis of E Protein

After analyzing the Blast results of E protein of four JEV isolates and the template reference given by SWISS-MODEL, four protein structural templates with the highest homology were selected. The PDB numbers are 3p54 (Method: X-RAY DIFFRACTION; Resolution: 2.10 Å), 5mv1 (Method: X-RAY DIFFRACTION; Resolution: 2.25 Å), 5mv2 (Method: X-RAY DIFFRACTION; Resolution: 2.10 Å) and 5wsn (Method: ELECTRON MICROSCOPY; Resolution: 4.30 Å), respectively. Moreover, 3p54 is the crystal structure of the E protein of the JEV vaccine strain obtained by X-ray diffraction. Finally, the subunit structure of the E protein of JEV CH/SA14-14-2MSV/2018 and four JEV isolates in their natural state was simulated and displayed by Modeller software. The tertiary structures of the above proteins were compared. The results showed only one amino acid difference in the tertiary structure between JEV CH/GD2011/2011 and CH/SA14-14-2MSV/2018. This variation site was at position 312, changing from K (lysine) to R (arginine). There was no significant difference in the overall structure between the two strains (Figure 8).

There were 11 amino acid differences between JEV CH/GD2015/2015 and CH/SA14-14-2MSV/2018 in the modeled 3D structures (Figure 9) and the variation sites were at the 107th, 129th, 138th, 176th~177th, 232, 244, 264, 279, 315 and 439 amino acids (Figure 10): F (phenylalanine) to L (leucine), T (threonine) to S (serine), K (lysine) to E (glutamic acid), V (valine) to I (isoleucine), A (alanine) to T (threonine), A (alanine) to T (threonine), G (glycine) to E (glutamic acid), H (histidine) mutated to L (leucine), M (methionine) mutated to K (lysine), V (valine) mutated to A (alanine), and R (arginine) mutated to K (lysine).

There were 13 amino acid differences in the modeled 3D structures between JEV CH/GD2018/2018, CH/GD2014/2014 and CH/SA14-14-2MSV/2018 (Figure 10). The variation sites were at 107, 129, 138, 176, 177, 222, 244, 264, 279, 315, 327, 366 and 439 amino acids: F (phenylalanine) to L (leucine), T (threonine) to M (methionine), K (lysine) to E (glutamic acid), V (valine) to I (isoleucine), A (alanine) to T (threonine), A (alanine) to S (serine), G (glycine) to E (glutamic acid), H (histidine) mutated to Q (glutamine), M (methionine) mutated to K (lysine), V (valine) mutated to A (alanine), S (serine) mutated to T (threonine), A (alanine) mutated to S (serine), and R (arginine) mutated to K (lysine).

## 4. Discussion

JE is an acute mosquito-borne infectious disease considered the leading cause of Asian viral encephalitis, of which about 50% occurs in China. Although the number of cases of JEV infection has decreased significantly since the launch of the national vaccination program in the 1970s, the number of reported cases remains high [6,33]. The prevalence of JE is related to climatic factors such as annual precipitation, and it is most common in southwestern and eastern China [7]. JEV can be replicated in mosquitoes and transmitted to pigs by hematophagy so that pigs can become the temporary host of JEV and the source of human infection [34]. In recent years, intensive pig farming has been accelerated, which may increase the probability of pigs being infected with JEV. In this study, 78 porcine JEV nucleic acids were detected in 263 samples collected from 14 different pig farms in southern China from 2011 to 2018, with a positive rate of 29.7%. Considering the limited number of pig farms collected in this survey and the limitations of collecting serum, brain tissue, and aborted stillbirth samples, this can still reflect that there is a certain degree of JEV natural infection in pigs in Guangdong, Guangxi, and other regions. Guangdong and Guangxi are located in the tropics and subtropics, with a warm and humid climate, high mosquito density, large number of domestic pigs, and natural conditions for JEV transmission and reproduction. Although there is no large-scale epidemic of Japanese encephalitis, its potential harm cannot be ignored.

Four strains of viruses were isolated from Vero cells to explore the biological characteristics of porcine JEV which were positive for JEV infection. Different cell culture and virus titer determination of four JEV isolates showed that four JEV isolates could proliferate stably in Vero cells, and the virus titer was as high as 10^8.5^ TCID 50/mL.

To understand the characteristics of the genetic evolution of JEV isolates, we determined the whole genome sequences of four JEV isolates in this study. In this study, we performed the phylogenetic analysis and genetic development of JEV by sequencing the entire genome nucleotide sequences of JEV and the nucleotide and amino acid sequences of the E protein encoded by the E gene. JEV is classified into GI, GII, GIII, GIV, and GV types. The JEV variants isolated in this study, CH/GD2011/2011 and CH/GD2015/2015, were classified as GIII at the whole genome nucleotide sequence, E gene nucleotide and amino acid sequence levels and the CH/GD2014/2014 and CH/GD2018/2018 were classified as GI. This is compatible with other studies stating that the predominant circulating strains were GIII and GI [3]. The insertion versus deletion analysis results of the amino acid sequence of each protein of JEV also corroborated the above conclusions. E protein variations, as an essential virulence protein of JEV, and related amino acid sites significantly impact the neurotoxicity and invasiveness of JEV. The current study shows that the amino acid sites involved in the virulence of the E protein include amino acids 107, 138, 123, 176, 177, 244, 279, 315 and 439; however, only the CH/GD2011/2011 harbored mutations that attenuated its neurotoxicity at amino acids 107, 176, 244 and 279. Specific amino acid mutations in the E protein are critical for viral virulence, including E107 and E138, which are JEV virulence regulatory genes; residue E107 is located in a highly conserved hairpin motif in the extended domain II. In contrast, this motif region contains a fusion peptide, and the mutation of E-107 can alter the fusion properties of E protein in cell culture [35]. Zheng et al. [28] demonstrated that the acidity/basicity of E138 residue affects JEV neurovirulence, and the mutation of essential residue in E138 contributes to the attenuation of neurovirulence. In contrast, the mutation of acidic residues enhanced the neurovirulence of the virus. Yang et al. [15] investigated the role of five amino acid residues (E-107, E-138, E-176, E-177, E-279) in the E protein in weakly virulent strains compared with the virulent parental strain, and the findings showed that amino acids at positions E-107 and E-138 played a critical role in the attenuation of neurovirulence in the CH/SA14-14-2MSV/2018 strain, within which E-107 (L→F) and E-138 (E→K) attenuated the neurotoxicity of CH/SA14-14-2MSV/2018. Notably, in the present study, the CH/GD2011/2011 conformed to the above mutations of the E protein at residues E-107 and E-138, providing a reference for subsequent studies on the pathogenicity of JEV isolates. Interestingly, the CH/GD2015/2015 consisted of methionine to threonine mutation at E-129, commonly found in GI or GIII type I to serine, as the cause of the mutation, and whether new mutations occurred in JEV awaits further confirmation. E-176 and E-177 are located in the main domain I of the E protein, a region rich in conformational epitopes sensitive to low pH. The area around the E-176 and E-177 clusters are also considered a virulence locus [36]. Zhou et al. [37] by obtaining the SCYA 2010901 strain after passaging the SCYA 201201 strain 120 times, compared the amino acid sequence of SCYA 2010901 with those of other JEV strains and found that although a single E-176 (I→R) mutation did not affect the viral growth in BHK-21 cells, it significantly reduced the neurotoxicity of JEV in vivo, and the four JEV isolates in this study, namely, E-176 (I→V) and E-177 (T→A), had no mutations in the remaining three. Except for E-176 and E-177, residue E-244 (E→G) and residue E-279 (K→M) also had the same mutation pattern in CH/GD2011/2011 vs. CH/SA14-14-2MSV/2018. E-244 is located in extended domain II and is an essential site outside the critical surveillance site for JE vaccine safety, and E-279 is located in a hinge region in the extended domain II β chain, which may play a regulatory role in E protein function. The present study’s analysis of amino acid genetic variations in the E protein revealed that the CH/GD2011/2011 strain, which shares most of the same amino acid mutation patterns as the vaccine strain, may become a successful vaccine strain for the prevention of JEV infection. 

The N-glycosylation of the protein plays an essential role in protein folding and can obtain the correct spatial conformation of the protein. Zhang [38] confirmed that the mutation of the N-glycosylation site N154 in the JEV E protein significantly enhanced the induced humoral immune response. Compared with the classical strains, the E protein of the four JEV isolates in this study was more conservative. Dechtawewat et al. [39] used immunoprecipitation with specific antibodies and liquid chromatography–tandem mass spectrometry (LC-MS/MS) to study the potential phosphorylation sites on the NS1 protein of dengue virus (DENV), revealing that DENV NS1 phosphorylation has a functional effect on viral amplification during DENV infection. These findings emphasize the importance of phosphorylation and can also be used as experimental evidence to verify the predicted phosphorylation sites in this study, which may pave the way for the future design of targeted specific antiviral drugs. The E protein is a vital antigen protein, and the results show that its phosphorylation site has high homology among different strains. It is suggested that the E protein of the four JEV isolates can perform its biological function typically. The changes in some E protein phosphorylation sites or N-glycosylation sites may be similar to the mutation of their amino acid sites, which can enhance or reduce the neurotoxicity of the virus strains.

Identifying antigenic epitopes provides a scientific basis for understanding disease aetiology, immune surveillance, developing diagnostic kits, and designing epitope-based subunit vaccines [40]. However, epitope identification is expensive and time-consuming because it requires the experimental screening of many potential epitope candidates. It is necessary to predict epitopes through computer prediction methods and tools for B and T cells. The software prediction found that the four JEV isolates had potential B cell epitopes at positions 24, 39, 183, 198, 332, 433 and 455, respectively. Compared with JEV CH/SA14-14-2MSV/2018, both CH/GD2014/2014 and CH/GD2018/2018 had a potential B cell epitope “LSLPWTSPSSTAWRNR” at 221–236, which provided a new insight for the screening of the JEV antigen epitope vaccine. The E protein of the four isolates all had dozens of T cell epitopes. It is worth noting that the confidence values of “GNYSAQVGASQAAKF” and “FLATGGVLVFLATNV” of the epitopes starting at 153,484 were above 30. The results show that the epitopes can be used to develop diagnostic kits and subunit vaccines based on epitopes in JEV.

Genes directly control biological traits by exploiting the structure of proteins. The molecular structure of a protein is very complex and can be divided into primary, secondary, tertiary and quaternary structures. Its secondary system can be used to further predict the tertiary and quaternary structure and to speculate the function of proteins [41,42]. Protein structural homology modeling provides a structural model for life science research when no experimental structure is available. The actual structural model should not only correctly represent the overall folding of a single protein chain but also correctly represent the atomic details of the interaction with essential cofactors and ligands [43]. In this study, the existing JEVE protein template was selected to simulate the E protein structure of the isolated JEV strain to show its three-dimensional structure and mutation site. The interaction between fusion loop (DII) and fusion loop pocket (DI-DIII) is particularly important for maintaining the JEV E dimer. The confirmed structure reveals the interaction between flavivirus’ fusion loop and fusion loop pocket, including the two amino acids in the fusion ring located in W101 and F108, respectively [44]. Interestingly, the W101 site of JEV E interacts with K312, N313, A315, and V323, while F108 interacts with A315 and D316 on the relative subunits [45]. In the amino acid mutation and tertiary structure of binding E protein, it was not difficult to see that the four JEV isolates did not mutate at E-101 or E-108. CH/GD2014/2014, CH/GD2015/2015, and CH/GD2018/2018 did not show mutations in E-312, E-313, E-315, E-316 or E-323. However, like CH/SA14-14-2MSV/2018, CH/GD2011/2011 mutated from A (alanine) to V (valine) in E-315. It is worth noting that CH/GD2011/2011 mutated from K (lysine) to R (arginine) in E-312.The protein modeled 3D structures of four JEV isolates was compared with that of the CH/SA14-14-2MSV/2018. Except that there is only one amino acid difference in modeled 3D structures between JEV CH/GD2011/2011 and CH/SA14-14-2MSV/2018: this variation site is at position 312, changing from K (lysine) to R (arginine). There are 11-13 amino acid differences between CH/GD2015/2015, CH/GD2014/2014, CH/GD2018/2018, and CH/SA14-14-2MSV/2018 with the evolution of the JEV virus. We do not know whether the existing effective JEV vaccine can effectively contain the spread of JEV in the future.

Through the prediction and analysis of glycosylation sites, phosphorylation sites, B cell epitopes, and T cell epitopes of the JEV E protein, as well as the study of the modeled 3D structures, we not only have a deeper understanding of the structure of JEV and E protein, but have also laid a foundation for the future study of the function and mechanisms of structural and nonstructural proteins. This can also lay the foundation for the research and design of more accurate and effective immunogenicity and antigenicity of the JE vaccine.

## Figures and Tables

**Figure 1 vaccines-10-01303-f001:**
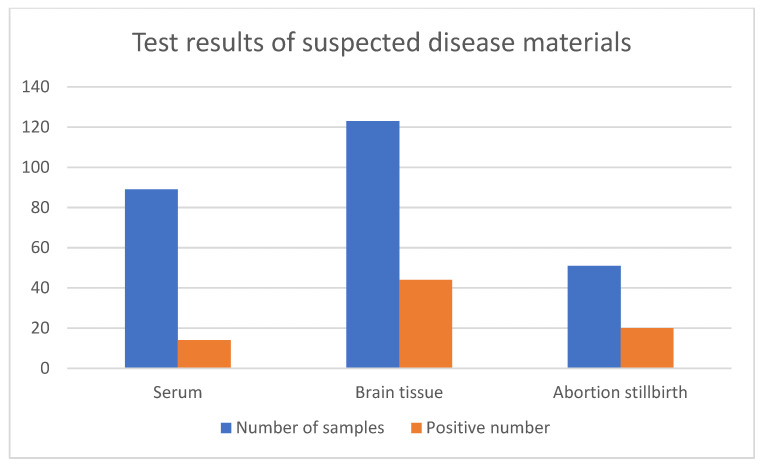
The results of detecting suspected JEV materials in South China from 2011 to 2018. Blue represents the total number of tests, and orange represents the number of positive samples.

**Figure 2 vaccines-10-01303-f002:**
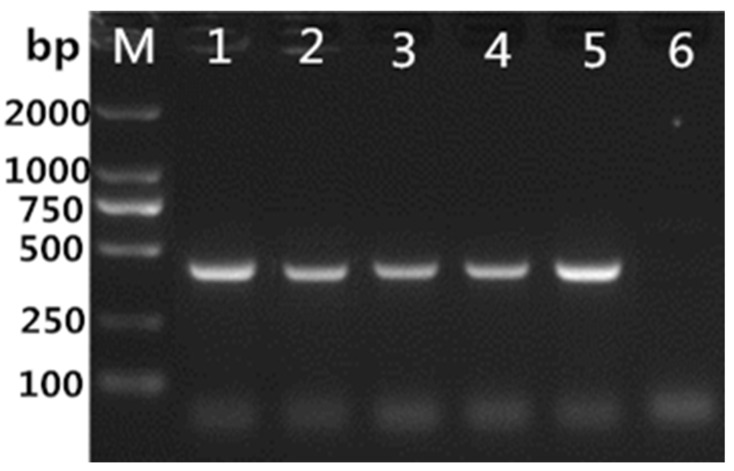
RT-PCR detection of JEV isolates. M: DL2000 DNA marker; 1: CH/GD2015/2015; 2: CH/GD2011/2011; 3: CH/GD2014/2014; 4: CH/GD2018/2018; 5: positive control; 6: negative control.

**Figure 3 vaccines-10-01303-f003:**
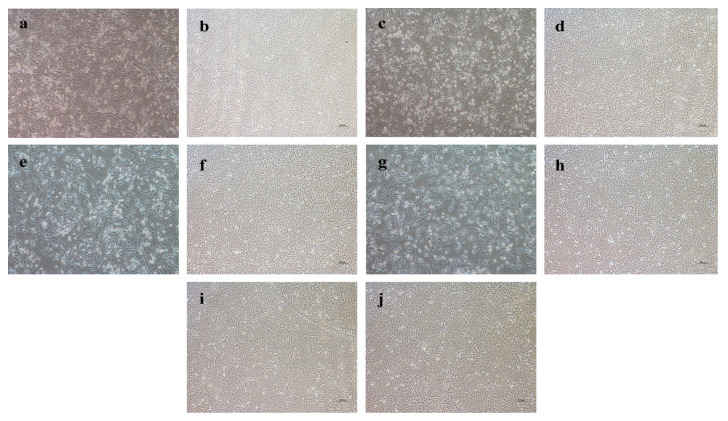
Identification of JEV isolates with specific serum. (**a**): CH/GD2011/2011 was inoculated; (**b**): CH/GD2011/2011 inoculation + positive serum treatment; (**c**): CH/GD2015/2015 was inoculated; (**d**): CH/GD2015/2015 inoculation + positive serum treatment; (**e**): CH/GD2014/2014 was inoculated; (**f**): CH/GD2014/2014 inoculation + positive serum treatment; (**g**): CH/GD2018/2018 was inoculated; (**h**): CH/GD2018/2018 inoculation + positive serum treatment; (**i**): positive serum control group; (**j**): blank control group (original magnification × 100).

**Figure 4 vaccines-10-01303-f004:**
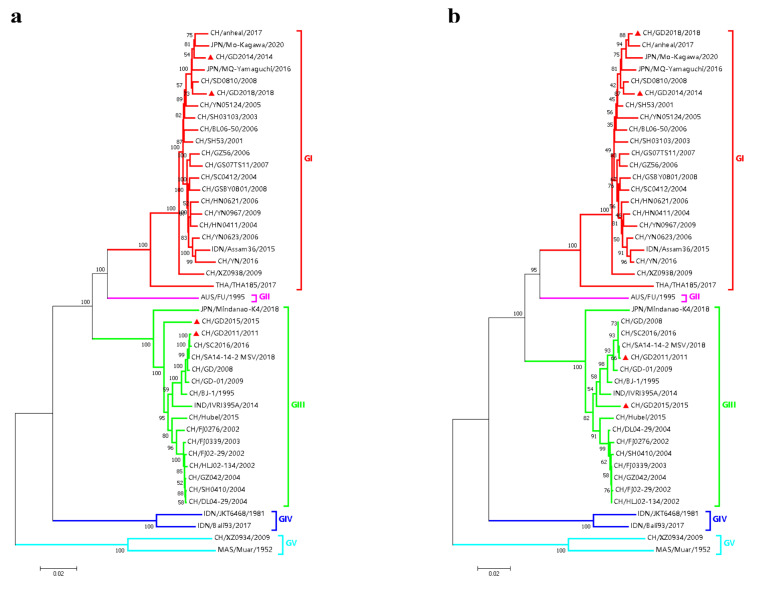
Evolutionary tree of JEV isolates. (**a**) Genomic phylogenetic relationship analysis of JEV isolates; (**b**) phylogenetic relationship analysis of the E gene of JEV isolates (the π in the picture is the isolate in this study.).

**Figure 5 vaccines-10-01303-f005:**
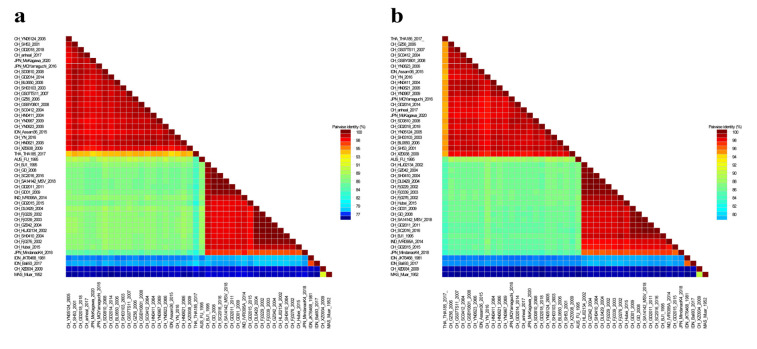
Nucleotide similarity of JEV isolates. (**a**) Nucleotide similarity of E gene of JEV isolates; (**b**) nucleotide similarity of JEV isolates.

**Figure 6 vaccines-10-01303-f006:**
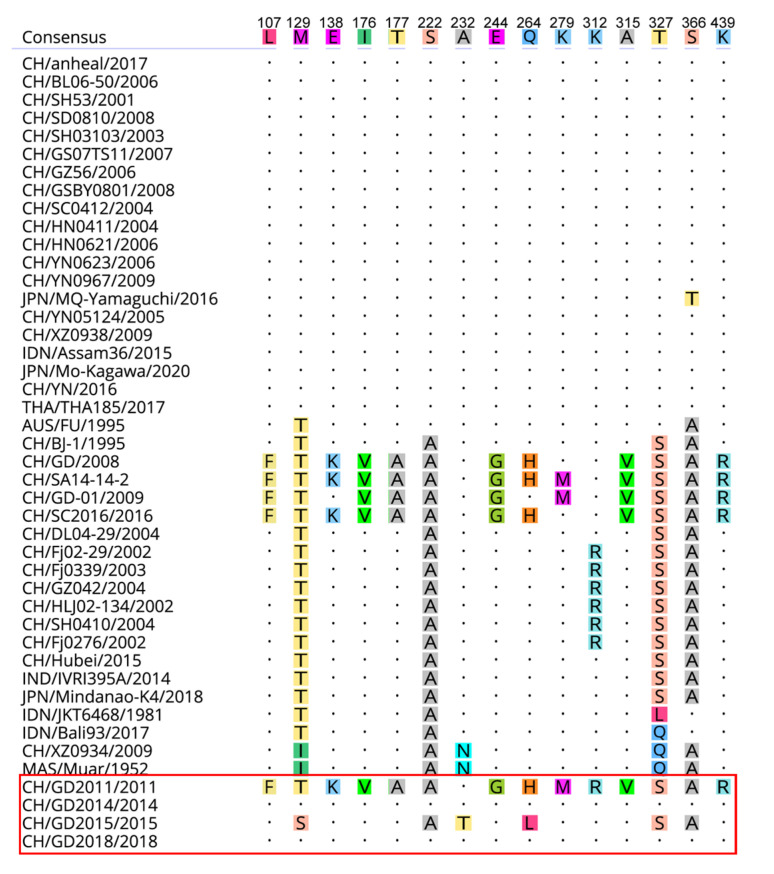
Amino acid mutation analysis of JEV E protein. Different amino acids are marked with different colors, and amino acids with the same position are replaced by “·”. (the isolates of this study are in the red box).

**Figure 7 vaccines-10-01303-f007:**
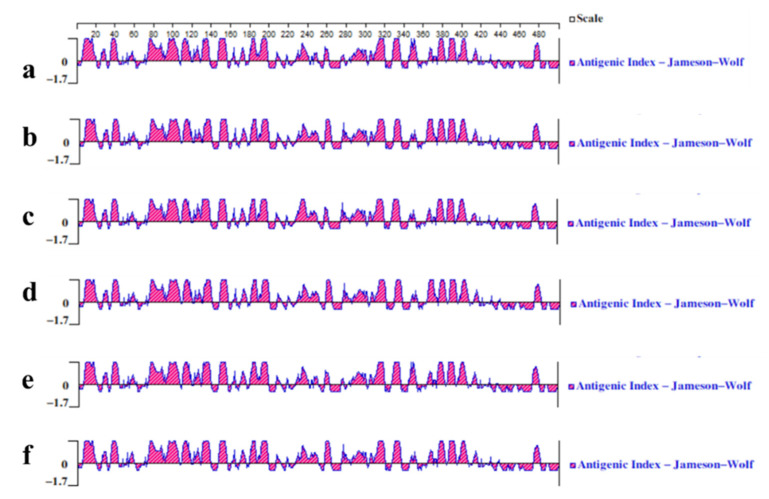
Antigenic index prediction of the E protein of JEV isolates. (**a**) CH/GD2011/2011; (**b**) CH/GD2015/2015; (**c**) CH/GD2014/2014; (**d**) CH/GD2018/2018; (**e**) CH/SA14-14-2 MSV/2018; and (**f**) CH/BJ-1/1995.

**Figure 8 vaccines-10-01303-f008:**
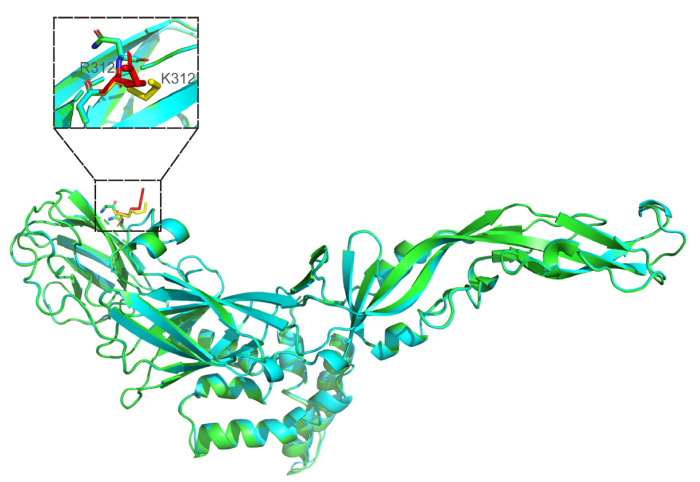
Display and comparison of modeled 3D structures between JEV CH/GD2011/2011 and CH/SA14-14-2 MSV/2018. The difference locus is at position 312 (green is the modeled 3D structures of vaccine strain; blue is the modeled 3D structures of CH/GD2011/2011; the different amino acids are distinguished by yellow and red: yellow for CH/SA14-14-2 MSV/2018 and red for CH/GD2011/2011.).

**Figure 9 vaccines-10-01303-f009:**
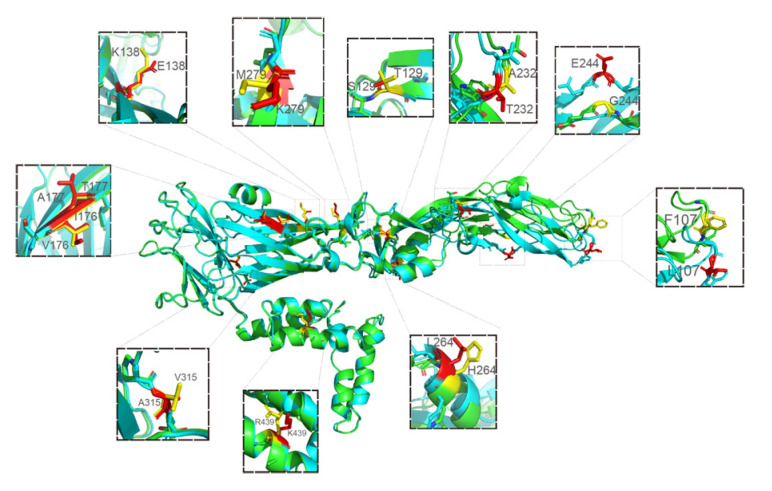
Display and comparison of modeled 3D structures between JEV CH/GD2015/2015 and CH/SA14-14-2 MSV/2018. The difference sites were at 107, 129, 138, 176, 177, 232, 244, 264, 279, 315 and 439 amino acids (green is the modeled 3D structures of vaccine strain; blue is the modeled 3D structures of CH/GD2015/2015; the different amino acids are distinguished by yellow and red: yellow for CH/SA14-14-2 MSV/2018 and red for CH/GD2015/2015.).

**Figure 10 vaccines-10-01303-f010:**
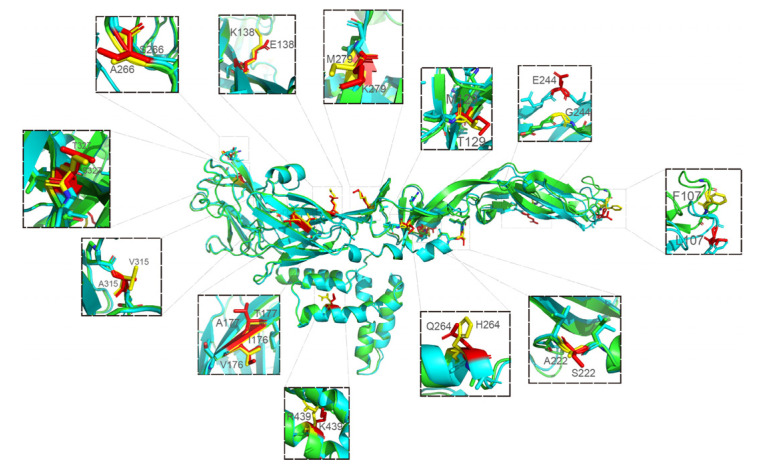
Display and comparison of modeled 3D structures between JEV CH/GD2014/2014 (CH/GD2018/2018) and CH/SA14-14-2 MSV/2018. The difference sites were at 107, 129, 138, 176, 177, 222, 244, 264, 279, 315, 327, 366 and 439 amino acids (green is the modeled 3D structures of vaccine strain; blue is the modeled 3D structures of CH/GD2014/2014 (CH/GD2018/2018); the different amino acids are distinguished by yellow and red: yellow for CH/SA14-14-2 MSV/2018 and red for CH/GD2014/2014 (CH/GD2018/2018).).

**Table 1 vaccines-10-01303-t001:** Primers for JEV gene amplification.

Primer	Sequence (5′~3′)	Product Length (bp)	Primer Position (nt)
JEV-P1-F	TATGCTGAAACGCGGCCTAC	2836	140~2975
JEV-P1-R	ACGGGTTGATGTGATGCCAA		
JEV-P2-F	GAGATATCGCTCAGCCCCAAA	2840	2762~5601
JEV-P2-R	GGGCATTTGAGTCGGGAAAAG		
JEV-P3-F	CCGCACGAGGATACATTGCT	2835	5494~8328
JEV-P3-R	CGTGATTGGAGTTTCGGGAC		
JEV-P4-F	TCTGCCCTTACATGCCCAAG	2716	8227~10,942
JEV-P4-R	GCTACATACTTCGGCGCTCT		

**Table 2 vaccines-10-01303-t002:** Common T-cell antigenic epitope sequence of JEV isolate E protein.

Number	T-Cell Epitope Sequence	The Starting Point of Amino Acid Position	Confidence Value
1	GNYSAQVGASQAAKF	153	34
2	FLATGGVLVFLATNV	484	32
3	HATKQSVVALGSQEG	246	28
4	GHGTVVIELSYSGSD	318	28
5	HALAGAIVVEYSSSV	264	27
6	DVRMINIEASQLAEV	42	26
7	EGGLHHALAGAIVVE	259	26
8	VEYSSSVMLTSGHLK	272	26
9	ARDRSIALAFLATGG	475	26
10	PCKIPIVSVASLNDM	334	25

## Data Availability

Not applicable.

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
