# Peer review of "Genomic Characteristics and E Protein Bioinformatics Analysis of JEV Isolates from South China from 2011 to 2018"

_vaccines, 2022, doi:10.3390/vaccines10081303_

Round 1
Reviewer 1 Report
The authors have investigated strains of the Japanese encephalitis virus (JEV) isolated between 2011 and 2018. JEV is a significant pathogen in livestock as well as humans. The JEV strains were sequenced and subjected to bioinformational analyses, including phylogenetic reconstructions, with focus on the mutations found in the E protein, which is directly involved in host cell receptor binding and cellular infection. The 3D structures of the isolates were modeled based on known X-ray and cryo EM structures of the JEV E protein.
Overall, the study looks at at various aspects related to the JEV E protein from the isolates but without any detailed focus. Where I would have expected to see a much more detailed discussion is on the structural location of the mutant positions and whether they might e.g. influence or not the viral structure or the likely interactions with the host receptors as described and postulated on in publications resulting from solved JEV structures.
Lines 75-76 and then throughout the text: The labeling of E protein positions as E107, E138 etc leads to confusion with standard amino acid labeling. The authors could add a parenthetical note to indicate the labeling scheme. Authors could mimic what others use: e.g. E-107 etc.
Phylogenetic analyses, methods and figure 4. Trees are presented and 1000 bootstraps were used to assess the robustness of the trees, yet there is no indication of these results on the trees.
Line 218: The genomic nucleotides, gene nucleotides, and deduced amino acid sequences … What is the difference between: genomic nucleotides and gene nucleotides ?
Is there any experimental evidence available that position N154 of the E protein is glycosylated? Line 260: “the confidence level reached 9gamma” ?
Line 268: “After phosphorylation, the protein has an electric charge, which changes the structure and causes a change in protein activity …” The protein also has a charge before phosphorylation, so the language should be corrected to indicate that phosphorylation adds an additional negative charge to the protein. Also, phosphorylation MAY cause a change in structure and/or molecular interactions.
The large number of predicted phosphorylation sites indicates the uncertainty in making such predictions. Taking into the account the structural context: exposed to solvent?, not buried in the dimer structure? location in the viral structure? might help identify some potential sites, but overall the prediction in not useful without other correlating data such as experimental evidence that can pinpoint what is a likely phosphorylation site.
Line 314: epitopesa ?
From line 316 onwards. Secondary structure predictions are not very useful here. Several structures of the E protein are known and provide the best data on the secondary structure. A few mutations may but are unlikely to cause any authentic structural alterations. Here, the purpose of the the predictions is questionable. Used to justify modeling?, but that should have been obvious since the E protein sequences are nearly identical.
Modeling structures: Very poorly described. References to the cited PDB codes 5mv1A, 5mv2A and 5wsnU are not provided. The last letter in each case is the chain label. Basic details on the structures (e.g. method used: cryo EM, X-ray diffraction? resolution?) are not provided.
I am not sure why modeling was even used given the nearly identical sequences. Mutations could have been introduced into the known structures. Nonetheless, modeling was used and is not well described in the methods. One would like to know all the manipulations made during the modeling process, for example energy minimization, which might later the structure. Only one structure referenced.
One would not expect differences except for mutated side chains. The following statement is obvious and the figure provides little added value. “There was no significant difference in the overall structure between the two strains(Figure 9). “
Since modeling was applied to the isolates, throughout the text they should be referred to as modeled 3D structures or as predicted structures.
Figure 11 is probably the most interesting figure in the manuscript but it is difficult to see what is taking place. Splitting into larger images would be useful. Adding labels to indicate the mutation e.g. F107F/T/M would be very useful instead of the complications in the text, i.e.: The variation sites are at 107, 129, 138, 176-177, 222, 244, 264, 279, 315, 327, 366 and 439 amino acids, respectively. F (phenylalanine) to L (leucine), T (threonine) to M (methionine), K (lysine) to E (glutamic acid), V (valine) to I (isoleucine), A (alanine) to T (threonine), A (alanine) to S (serine), G (glycine) to E (glutamic acid), H (histidine) mutated to Q (gluta- mine), M (methionine) mutated to K (lysine), V (valine) mutated to A (alanine), S (serine) mutated to T (threonine), A (alanine) mutated to S (serine) and R (arginine) mutated to K (lysine).
The context with respect to the dimer and to the intact virus would have also been useful. The published structure papers and cited references are a good resource for describing possible effects or not due to the mutations seen in the isolates.
Line 65. bands (?) -> strands or β-strands
Author Response
Dear reviewer:
Thank you for your valuable comments and suggestions. Based on your comments and suggestions, we make the following changes:
response:
Figures 8, 9 and 10 have been modified in this manuscript to show the structural location of the mutation in the E protein, and the structural location of the mutation has been supplemented in the Discussion section.
Lines 73-74 and then throughout the text - "E107, E138" et al. was changed to " E-107, E-138" et al.
The evolutionary tree of JEV isolates in Figure 4 has been modified, and the data for 1000 bootstraps assessment of stability can be seen in Figure 4.
Line 228 "gene nucleotides"deleted.
Experiments have shown that the N154 position of the E protein is glycosylated, and the amino acid mutation of the E protein combined in this study also shows the conservative type of N154.
Lines 277-283 The protein does also carry a charge before phosphorylation, we have modified. At the same time, we also added " the unique size and charge characteristics of covalently linked phosphates also allow phosphorylated proteins to specifically recognize phosphorylated proteins through phosphorylation-specific binding domains in other proteins, thus promoting inducible protein-protein interactions.".
Lines 487-496 The experimental evidence of the phosphorylation sites of dengue virus has been identified, which can be used for reference in the prediction of phosphorylation sites in this study.
Line 329 The word "epitopesa" was changed to "epitope sequence".
All secondary structures involved have been deleted in this manuscript.
Lines 349-352 Supplements method and resolution of PDB codes used for modeling structures.
This study selected the existing JEV E protein template to simulate the structure of the isolated JEV strain E protein, aiming to display its three-dimensional structure and mutation sites. This is an attempt.
Modifications have been made to Figure 9, including the labeling of mutation locations.
The "tertiary structure" that appears in the full text has been changed to "modeled 3D structures".
Modifications have been made to Figure 11, and add labels to indicate mutations.
More background on dimers and viruses has been added in the Discussion and Preface.
Lines 75-76 It has now been revised due to errors in the layout of the foreword in the manuscript.
Thank you again for your responsible and detailed suggestions.
Reviewer 2 Report
Manuscript titled "Genomic characteristics and E protein bioinformatics analysis of JEV isolates from South China from 2011 to 2018" by Sun et al is a nice piece of work. Overall the draft is in nice form but need minor improvement before it can be accepted for publication.
Some of the comments to authors are
1) Minor improvement required in writing part.
2) better to replace the word strain with variant as term strain is not suitable for virus and variant is more appropriate.
3) In case of fig. 3 put more zoom in image and of high resolution. Difficult to understand
4) Same goes with fig.5 and fig.8
5) whether % of positives in serum, brain tissue and abortion still birth is expected or normal?. Author should discuss this in text.
Author Response
Dear reviewer:
Thank you for your valuable comments and suggestions. Based on your comments and suggestions, we make the following changes:
response:
1.The full text writing section has been revised.
2.Line 25、Line 171、Line 211、Line 218、Line 220 et al. - The word "strain" was changed to "variant".
3.Fig. 3 three has been replaced with a higher resolution image.
4.Fig. 5 three has been replaced with a higher resolution image ; Fig. 8 refers to protein secondary structure, which has been removed at the suggestion of another reviewer.
5.The percent positive for detected samples has been discussed in the Discussion section.
Thank you again for your responsible and detailed suggestions.